# FutureFill: Fast Generation from Convolutional Sequence Models

## Abstract

We address the challenge of efficient auto-regressive generation in sequence prediction models by introducing FutureFill—a method for fast generation that applies to any sequence prediction algorithm based on convolutional operators. Our approach reduces the generation time requirement from quadratic to quasilinear relative to the context length. Additionally, FutureFill requires a prefill cache sized only by the number of tokens generated, which is smaller than the cache requirements for standard convolutional and attention-based models. We validate our theoretical findings with experimental evidence demonstrating correctness and efficiency gains in a synthetic generation task.

## 1 Introduction

Large Transformer models Vaswani et al. (2017) have become the method of choice for sequence prediction tasks such as language modeling and machine translation. Despite their success, they face a key computational limitation: the attention mechanism, their core innovation, incurs a quadratic computational cost during training and inference. This inefficiency has spurred interest in alternative architectures that can handle long sequences more efficiently.

Convolution-based sequence prediction models Li et al. (2022); Poli et al. (2023); Agarwal et al. (2023); Fu et al. (2024) have emerged as strong contenders, primarily due to their ability to leverage the Fast Fourier Transform (FFT) for near-linear scaling with sequence length during training. These models build upon the advancements in State Space Models (SSMs), which have shown promise in modeling long sequences across diverse modalities Gu et al. (2021a); Dao et al. (2022); Gupta et al. (2022); Orvieto et al. (2023); Poli et al. (2023); Gu & Dao (2023). Convolutional models offer a more general framework than SSMs because they can represent any linear dynamical system (LDS) without being constrained by the dimensionality of hidden states Agarwal et al. (2023). This flexibility has led to recent developments that theoretically and empirically handle longer contexts more effectively. Notable among these are Spectral State Space Models or Spectral Transform Units (STUs) Agarwal et al. (2023), which use spectral filtering algorithms Hazan et al. (2017; 2018) to transform inputs into better-conditioned bases for long-term memory. Another approach is Hyena Poli et al. (2023), which learns implicitly parameterized Markov operators. Both methods exploit the duality between time-domain convolution and frequency-domain multiplication to accelerate prediction via the FFT.

While SSMs and recurrent models benefit from fast inference times independent of sequence length, making them attractive for large-scale language modeling, convolutional models have been hindered by slower token generation during inference. The best-known result for generating tokens with convolutional models is quadratic in sequence length—comparable to attention-based models (see Massaroli et al. (2024) Lemma 2.1). This limitation has prompted research into distilling state-space models from convolutional models Massaroli et al. (2024), but such approximations lack comprehensive understanding regarding their approximation gaps due to the broader representational capacity of convolutional models.

In this paper, we address the problem of exact auto-regressive generation from given convolutional models, significantly improving both the generation time and cache size requirements. We present our main results in two settings:

1. **Generation from Scratch:** When generating $L$ tokens from scratch, we demonstrate that long convolutional sequence predictors can generate these tokens in total time $O(L \log^2 L)$ with total memory $O(L)$. This improves upon previous methods that require $O(L^2)$ time for generation. We further provide a memory-efficient version wherein the total runtime increases to $O(L^{3/2} \sqrt{\log(L)})$ but the memory requirement is bounded by $O(\sqrt{L \log L})$.

2. **Generation with Prompt:** When generating $K$ tokens starting from a prompt of length $L$, we show that the total generation time is $O(L \log L + K \log^2 K)$ with a cache size requirement of $O(K)$. Previously, the best-known requirements for convolutional models were a total generation time bounded by $O(L \log L + LK + K^2)$ and a cache size bounded by $O(L)$ (Massaroli et al., 2024).

Importantly, our results pertain to provably exact generation from convolutional models without relying on any approximations. Moreover, our methods are applicable to any convolutional model, regardless of how it was trained. The following table compares our algorithm with a standard exact implementation of convolution. We also provide a comparison of the time and cache size requirements for exact computation in attention-based models.

| Method | Runtime | Memory |
|---|---|---|
| Standard Conv | $L^2$ | $L$ |
| Standard Attn. | $L^2$ | $L$ |
| EpochedFF (ours) | $L^{3/2}\sqrt{\log L}$ | $\sqrt{L \log L}$ |
| ContinuousFF (ours) | $L \log^2 L$ | $L$ |

| Prefill+Genertation Runtime | Generation Cache Size |
|---|---|
| $LK + L \log L + K^2$ | $L$ |
| $L^2 + KL$ | $L$ |
| $L \log L + K^{3/2}\sqrt{\log K}$ | $K$ |
| $L \log L + K \log^2 K$ | $K$ |

(a) Comparison for generating $L$ tokens from scratch. Runtime is in asymptotic notation, i.e. $O(\cdot)$ is omitted for brevity.

(b) Comparison for generating $K$ tokens starting from a prompt of length $L$, runtime and cache-size are in asymptotic notation, i.e. $O(\cdot)$ is omitted for brevity.

Our results for generation from convolutional models are based on building efficient algorithms for an online version of the problem of computing convolutions. In this problem, the algorithm is tasked to compute the convolution of two sequences $u * \phi$, however the challenge is to release iteratively at time $t$ the value of $[u * \phi]_t$, where the sequence $\phi$ is fully available to the algorithm but the sequence $u$ streams in one-coordinate at a time.

While the FFT algorithm allows for an $O(L \log L)$-time *offline* algorithm for the convolution of two $L$-length sequences, whether a similar result exists for the online model was not known. Naively, since $[u * \phi]_t = \langle u_{1:t}, \phi_{t:1} \rangle$, the total output can be computed in time $O(L^2)$. In this paper we demonstrate using repeated calls to appropriately constructed FFT-subroutines to compute the *future* effect of past tokens (a routine we call FutureFill), one can compute the convolution in the online model with a total computational complexity of $O(L \log^2(L))$, nearly matching its offline counterpart and significantly improving over the naive algorithm which was the best known (Massaroli et al., 2024).

It is worth noting that the naive algorithm for computing online convolution, albeit slow, does not require any additional memory other than the memory used for storing the sequences $v, w$. Such memory is often a bottleneck in practical sequence generation settings and is referred to as the size of the generation cache. For context the size of the generation cache for attention models is $O(L)$, i.e. proportional to the length of the prefill-context and the generation length. We further show that when generating from convolutional models, one can construct a trade-off for the computational complexity (i.e. flops) and memory (i.e. generation cache size) using the FutureFill sub-routine. We highlight two points on this trade-off spectrum via two algorithmic setups both employing FutureFill. We detail this trade-off in Table 1.

## 1.1 RELATED WORK

**State space models and convolutional sequence prediction.** Recurrent neural networks have been revisited in the recent deep learning literature for sequential prediction in the form of state space models (SSM), many of whom can be parameterized as convolutional models. Gu et al. (2020)

| Algorithm | Computational Complexity (Flops) | Memory (Generation Cache-Size) |
|---|---|---|
| Naive | $O(L^2)$ | O(1) |
| Epoched-FutureFill (ours) | $O(L^{3/2} \log L)$ | $O(\sqrt{L})$ |
| Continuous-FutureFill (ours) | $O(L \log^2 L)$ | $O(L)$ |

Table 1: Comparison of results for Online convolution.

propose the HiPPO framework for continuous-time memorization, and shows that with a special class of system matrices $A$ (HiPPO matrices), SSMs have the capacity for long-range memory. Later work Gu et al. (2021b;a); Gupta et al. (2022); Smith et al. (2023) focus on removing nonlinearities and devising computationally efficient methods that are also numerically stable. To improve the performance of SSMs on language modeling tasks Dao et al. (2022) propose architectural changes as well as faster FFT algorithms with better hardware utilization, to close the speed gap between SSMs and Transformers. Further investigation in Orvieto et al. (2023) shows that training SSM is brittle in terms of various hyperparameters. Various convolutional models have been proposed for sequence modelling, see e.g. Fu et al. (2023); Li et al. (2022); Shi et al. (2023a). These papers parameterize the convolution kernels with specific structures. The Hyena architecture was proposed in Poli et al. (2023) and distilling it into a SSM was studied in Massaroli et al. (2024). Other studies in convolutional models include LongConv Fu et al. (2023) and SGConv Li et al. (2022) architectures, as well as multi-resolution convolutional models Shi et al. (2023b).

**Spectral filtering.** A promising technique for learning in linear dynamical systems with long memory is called spectral filtering put forth in Hazan et al. (2017). This work studies online prediction of the sequence of observations $y_t$, and the goal is to predict as well as the best symmetric LDS using past inputs and observations. Directly learning the dynamics is a non-convex optimization problem, and spectral filtering is developed as an improper learning technique with an efficient, polynomial-time algorithm and near-optimal regret guarantees. Different from regression-based methods that aim to identify the system dynamics, spectral filtering's guarantee does not depend on the stability of the underlying system, and is the first method to obtain condition number-free regret guarantees for the MIMO setting. Extension to asymmetric dynamical systems was further studied in Hazan et al. (2018). Spectral filtering is particularly relevant to this study since it is a convolutional model with fixed filters. Thus, our results immidiately apply to this technique and imply provable regret bounds with guaranteed running time bounds in the online learning model which improve upon state of the art.

**Online learning and regret minimization in sequence prediction.** The methodology of online convex optimization, see e.g. Hazan et al. (2016), applies to sequences prediction naturally. In this setting, a learner iteratively predicts, and suffers a loss according to an adversarially chosen loss function. Since nature is assumed to be adversarial, statistical guarantees are not applicable, and performance is measured in terms of regret, or the difference between the total loss and that of the best algorithm in hindsight from a class of predictors. This is a particulary useful setting for sequential prediction since no assumption about the sequence is made, and it leads to robust methods. Sequential prediction methods that apply to dynamical systems are more complex as they incorporate the notion of a state. Recently the theory of online convex optimization has been applied to learning in dynamical systems, and in this context, the spectral filtering methodology was devised. See Hazan & Singh (2022) for an introduction to this area.

## 2 SETTING

### 2.1 ONLINE CONVOLUTIONS

**Notation:** For an input sequence $\{u_t\}$ we denote by $u_{1:t}$ the sequence of inputs $u_1, ..., u_t$. For any $i \leq j$ let $u_{i:j}$ denote the sub-sequence $u_i, u_{i+1}, \ldots u_j$. When $i > j$, $u_{i:j}$ denotes the subsequence $u_{j:i}$ in reverse order. Thus $u_{t:1}$ represents the sequence in reverse order. We also denote $[k] = \{1, 2, ..., k\}$ as a set of $k$ natural numbers. Given a multi-dimensional sequence $u_1 \ldots u_t$ where each $u_i \in \mathbb{R}^d$ and given a vector $v \in \mathbb{R}^t$, for brevity of notation we overload the definition of

inner products by defining $y = \langle v, u_{1:t} \rangle$ with $y \in \mathbb{R}^d$ as $y_j = \sum_{i=1}^t v_i \cdot [u_i]_j \in \mathbb{R}$. That is the inner-product along the time dimension is applied on each input dimension separately.

**Convolution:** The convolution operator between two vectors $u, \phi \in \mathbb{R}^t$ outputs a sequence of length $t$ whose element at any position $s \in [t]$ [1] is defined as

$$[u * \phi](s) = \sum_{i=1}^s u_i \phi_{s+1-i} = \langle u_{1:s}, \phi_{s:1} \rangle. \tag{1}$$

A classical result in the theory of algorithms is that given two vectors $u, \phi \in \mathbb{R}^t$, their convolution can be computed in time $O(t \log t)$, using the FFT algorithm.

**Online Convolution:** We consider the problem of performing the convolution $u * \phi$ when one of the sequences $\phi$ is fully available to the model, however the other sequence $u$ *streams* in, i.e. the element $u_t$ is made available to the model at the start of round $t$, at which point it is expected to release the output $[u * \phi]_t$. This model of online convolution is immediately relevant to the online auto-regressive generation of tokens from a convolutional sequence model as the output token at time $t$ becomes the input for the next round and hence is only available post generation. In this setting, the sequence $u$ corresponds to generated tokens and the sequence $\phi$ corresponds to the convolutional filter which the model has full access to. We further detail the setup of sequence generation in the next subsection.

## 2.2 SEQUENCE PREDICTION:

In **sequence prediction**, the input is a sequence of tokens denoted $u_1, ..., u_t, ...$, where $u_t \in \mathbb{R}^{d_{in}}$. The predictor's task is to generate a sequence $\hat{y}_1, ..., \hat{y}_t, ...$, where $\hat{y}_t \in \mathbb{R}^{d_{out}}$ is generated after observing $u_1, ..., u_{t-1}$. The output $y_t$ is observed after the predictor generates $\hat{y}_t$. The quality of the prediction is measured by the distance between the predicted and observed outputs according to a loss function $\ell_t(\hat{y}_t, y_t)$, for example the mean square error $\|\hat{y}_t - y_t\|^2$.

## 2.3 ONLINE SEQUENCE PREDICTION

In the **online sequence prediction** setting, an online learner iteratively sees an input $u_t$ and has to predict output $\hat{y}_t$, after which the true output $y_t$ is revealed. The goal is to minimize error according to a given Lipschitz loss function $\ell_t(y_t, \hat{y}_t)$. In online learning it is uncommon to assume that the true sequence was generated by the same family of models as those learned by the learner. For this reason the metric of performance is taken to be regret. Given a class of possible predictors, the goal is to minimize regret w.r.t. these predictors. For example, a linear predictor predicts according to the rule

$$\pi_{M_{1:k}, N_{1:l}}(u_{1:t}, y_{1:t-1}) = \sum_{i=1}^k M_i u_{t-i} + \sum_{j=1}^l N_j y_{t-j}.$$

The performance of a prediction algorithm $\mathcal{A}$ is measured by regret, or difference in total loss, vs. a class of predictors $\prod$, such as that of linear predictors, e.g.

$$\text{Regret}_T(\mathcal{A}) = \sum_{t=1}^T \ell_t(y_t, \hat{y}_t^{\mathcal{A}}) - \min_{\pi \in \prod} \sum_{t=1}^T \ell_t(y_t, \hat{y}_t^\pi).$$

This formulation is valid for online sequence prediction of any signal. We are particularly interested in signals that are generated by dynamical systems. A time-invariant linear dynamical system is given by the dynamics equations

$$x_{t+1} = Ax_t + Bu_t + w_t, \ \ y_t = Cx_t + Du_t + \zeta_t,$$

where $x_t$ is the (hidden) state, $u_t$ is the input or control to the system, and $y_t$ is the observation. The terms $w_t, \zeta_t$ are noise terms, and the matrices $A, B, C, D$ are called the system matrices. A linear

---

[1]This definition corresponds to the *valid* mode of convolution in typical implementations of convolution e.g. scipy.

dynamical predictor with parameters $A, B, C, D$ predicts according to

$$\pi_{ABCD}(u_{1:t}, y_{1:t-1}) = \sum_{i=1}^{t-1} CA^{i-1}Bu_{t-i} + Du_t.$$

The best such predictor for a given sequence is also called the optimal open loop predictor, and it is accurate if the signal is generated by a LDS without noise.

## 2.4 AUTO-REGRESSIVE SEQUENCE GENERATION FROM A PROMPT

Another mode of sequence prediction with large language models being its core use-case is that of auto-regressive sequence generation starting from a prompt. Herein the sequence model has to generate a specified number of tokens given a certain context. This is depicted in Figure 1. The setting of auto-regressive generation from a prompt consists of two stages, the prefill stage and the decode stage. During the prefill stage, the model ingests the context vector and generates a cache that stores information required in the decode stage.

In the decode stage, the model takes the cache and the most recently generated token as input and generates the next output token. Then the cache is updated with the most recent input token. We denote the generation length at the decode stage with $K$. In contrast to pre-training, where the model takes in a training sequence and predicts the next token, in the prefill generation setting the model only has access to the cache and the most recent token when making a prediction.

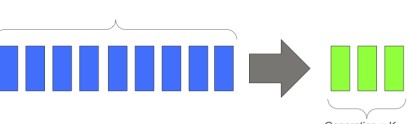

Figure 1: Auto-regressive sequence generation from a prompt.

## 2.5 ABSTRACTING CONVOLUTIONAL SEQUENCE PREDICTION

We define a convolutional sequence prediction model to be given by a *filter*, which is a vector denoted by $\phi \in \mathbb{R}^L$ where $L$ is considered the *context length* of the model. It takes as an input a sequence $u$, and outputs a prediction sequence. The above definition can be extended to multiple filter *channels* and nonlinearities, as we elaborate below with different examples. Formally, a single output in the predicted sequence using a convolutional sequence model is given by

$$\hat{y}_t = \langle \phi, u_{t:t-L} \rangle. \tag{2}$$

This paradigm captures several prominent convolutional sequence models considered in the literature. We highlight some of them below. The online convolution technique proposed by us can be used with all the models below in straightforward manner leading to generation time improvement from $O(L^2)$ to $O(L\log^2 L)$.

**State Space Models**   Discrete state space models such as those considered in Gu et al. (2021a) have shown considerable success/adoption for long range sequence modelling. A typical state space model can be defined via the following definition of a Linear Dynamical System (LDS)

$$x_t = Ax_{t-1} + Bu_t, y_t = Cx_t + Du_t \tag{3}$$

where $u, y$ are the input and output sequences and $A, B, C, D$ are the learned parameters. Various papers deal with specifications of this model including prescriptions for initialization (Gu et al., 2020), diagonal versions (Gupta et al., 2022), gating (Mehta et al., 2023) and other effective simplifications (Smith et al., 2023). All these models can be captured via a convolutional model by noticing that the output sequence $y$ in (3) can be written as

$$y = \phi * u + Du$$

where the filter $\phi$ takes the form $\phi_i = CA^{i-1}B$. Thus a convolutional sequential model with learnable filters $\phi$ generalizes these state space models. However, SSM are more efficient for generation and require only constant time for generating a token, where the constant depends on the size of the SSM representation.

**LongConv/SGConv.** The LongConv (Fu et al., 2023) and SGConv (Li et al., 2022) architectures, exploit the above connection and propose direct regularizations of the kernel to bias them towards kernels representing a state space model.

**Spectral Transform Units.** The STU architecture was proposed in Agarwal et al. (2023) based on the technique of spectral filtering for linear dynamical systems (Hazan et al., 2017; 2018). These are basically convolutional sequence models based on carefully constructed filters that are **not data dependent**. Rather, let $\phi_1, ..., \phi_k$ be the first $k$ eigenvectors of the Hankel matrix $H_L$ given by

$$H_L = \int_0^1 \mu_\alpha \mu_\alpha^\top d\alpha \ \in \mathbb{R}^{L \times L}, \ \mu_\alpha = (\alpha - 1)[1, \alpha, \alpha^2, .., \alpha^{L-1}].$$

Then the STU outputs a prediction according to the following rule [2] $\hat{y}_t = \sum_{i=1}^k M_i \langle \phi_i, u_{t:t-L} \rangle$, where $\phi_i$ are the eigenvectors as above and $M_{1:k}$ are learned projection matrices. The STU architecture is particularly appealing for learning in dynamical systems with long context, as it has theoretical guarantees for this setting, as spelled out in Agarwal et al. (2023).

**Hyena.** The Hyena architecture proposed in Poli et al. (2023), sequentially applies convolutions and element-wise products alternately. Formally, given an input $u_{1:T}$, $N + 1$ linear projections $v, x_1, \ldots x_N$ of the input are constructed (similar to the $q, k, v$ sequence in self-attention). The hyena operator as a sequence of convolution with learnable filters $h_1 \ldots h_N$ is then given by

$$y = x^N \cdot \left( h^N * \left( x^{N-1} \cdot \left( h^{N-1} * (\ldots) \right) \right) \right).$$

## 3 EFFICIENT ONLINE CONVOLUTIONS USING FUTUREFILL

We begin by introducing a simple convenient primitive which we call FutureFill, which forms the crucial building block of our algorithms. Intuitively FutureFill corresponds to computing the *contribution* of the current and previously generated tokens on the future tokens yet to be generated. For a convolutional model (and unlike attention) this contribution can be efficiently determined without even having generated the future tokens. Here onwards, for brevity of notation for any $v \in \mathbb{R}^t$, we assume $v_j = 0$ for any $j \leq 0$ or any $j > t$. Formally, given two sequences $v \in \mathbb{R}^{t_1}$, $w \in \mathbb{R}^{t_2}$ we define FutureFill$(v, w) \in \mathbb{R}^{t_2 - 1}$ as [3]

$$\forall s \in [t_2 - 1] \ \ [\text{FutureFill}(v, w)]_s = \sum_{i=1}^{t_2 - s} v_{t_1 - i + 1} \cdot w_{s+i}.$$

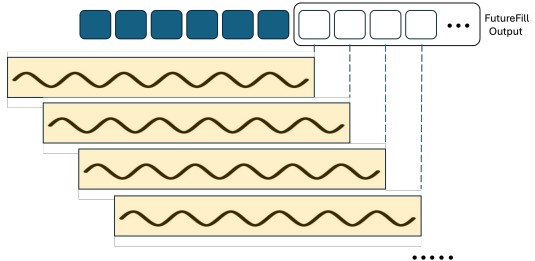

Figure 2: FutureFill operation between an input sequence and a convolutional filter.

Figure 2 depicts the FutureFill operation between an input sequence and a convolutional filter. Conceptually, $[\text{FutureFill}(v, w)]_s$ is the contribution of the input $v$ of length $t_1$ to the output $[v * w]$ at position $t_1 + s$. The FFT algorithm for convolutions can easily be extended to compute the Future-Fill as well in time at most $O((t_1 + t_2) \log(t_1 + t_2))$. For example the *full* mode of a standard conv

---

[2] more precisely, there are additional linear and constant terms depending on the exact filters used, such as $\hat{y}_t = \hat{y}_{t-2} + \sum_{i=1}^3 M_i^u u_{t+1-i} + \sum_{i=1}^k M_i \langle \phi_i, u_{t:t-L} \rangle$, see Agarwal et al. (2023) for more details.

[3] recall that we denote $[x] = \{1 \ldots x\}$.

implementation (e.g. scipy) can be used to compute FutureFill in the following way under Python slicing convention (exclusive of the last index)

```
FutureFill(v, w) = scipy.linalg.conv(v, w, mode=full)[t_1:t_1+t_2-1]
```

To leverage $FutureFill$ into an efficient way to generate tokens from a convolutional model, consider the following simple proposition that follows from the definition of convolution.

**Proposition 1.** *Given two vectors $a, b \in \mathbb{R}^t$, we have that*

$$\forall t_1, s \in [t] \qquad [a * b]_s = \begin{cases} [a_{1:t_1} * b_{1:t_1}]_s & \text{if } s \leq t_1 \\ [a_{t_1+1:t} * b_{1:t-t_1}]_{s-t_1} + [\text{FutureFill}(a_{1:t_1}, b)]_{s-t_1} & \text{otherwise} \end{cases}$$

We provide a proof of the proposition in the appendix. We use the above proposition to design efficient algorithms for online convolution.

### 3.1 EPOCHED-FUTUREFILL: EFFICIENT ONLINE CONVOLUTIONAL PREDICTION

When computing online convolutions, the FutureFill routine allows for the efficient pre-computation for the effect of past tokens on future tokens. We leverage this property towards online convolution via the Epoched-FutureFill procedure outlined in Algorithm 1.

---
**Algorithm 1** Epoched-FutureFill: Efficient Online Convolutional Prediction
---
1: **Input:** Convolutional filter $\phi \in \mathbb{R}^L$. Input sequence $u \in \mathbb{R}^L$, streaming one coordinate every round. $K$, the epoch length.
2: Set $\tau = 1$. Set FutureFill cache $C \in \mathbb{R}^K$ to 0.
3: **for** $t = 1, 2, ..., L$ **do**
4:     Receive $u_t$.
5:     Compute and output    $\hat{y}_t = \sum_{j=1}^{\tau} u_{t+1-j} \cdot \phi_j + C_\tau$
6:     **if** $\tau = K$ **then**
7:         Compute FutureFill cache $C \in \mathbb{R}^K$ defined as $C_j = [\text{FutureFill}(u_{1:t}, \phi_{1:t+K})]_j$.
8:         $\tau \leftarrow 1$
9:     **else**
10:         $\tau \leftarrow \tau + 1$
11:     **end if**
12: **end for**
---

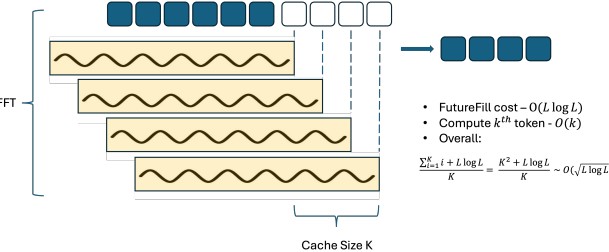

Figure 3: Illustration for Algorithm 1

In the following lemma we state and prove the properties that Epoched-FutureFill enjoys. The theorem provides a trade-off between the additional memory overhead and total runtime incurred by the algorithm. In particular, the runtime in this tradeoff is optimized when the total memory is $O(\sqrt{L \log L})$ leading to a total runtime of $O(L^{3/2}\sqrt{\log L})$.

**Theorem 2.** *Algorithm 1 computes the online convolution of sequences with length $L$ and runs in total time $O\left(\frac{L^2 \log L}{K} + KL\right)$ with a total additional memory requirement of $O(K)$. In particular setting $K = \sqrt{L \log L}$, we get that Algorithm 1 computes online convolution in $O(L^{3/2}\sqrt{\log L})$ total time and $O(\sqrt{L \log L})$ memory.*

*Proof.* Since the proof of correctness is mainly careful accounting the contributions for various indices, we provide it in the appendix. We prove the running time bounds below. The running time consists of two components as follows:

1. Every iteration, line 5 is executed. One term, $C_\tau$, has already been computed and saved in line 7. We can retrieve it in time $O(1)$. The other term is a sum of $\tau$ products, which can be computed in time $\tau$.

2. Every $K$ iterations, we execute line 7 and update the terms in the cache. The FutureFill operation can be computed via the FFT taking at most $O(L \log L)$ time.

The overall running time is computed by summing over the $L$ iterations. In each block of $K$ iterations, we apply FFT exactly once, and hence the total computational complexity is

$$\frac{L}{K} \left( L \log L + \sum_{\tau=1}^{K} \tau \right) = O \left( \frac{L^2 \log L}{K} + KL \right) = O \left( L^{3/2} \sqrt{\log L} \right),$$

where the last equality holds when the cache size $K$ is chosen to minimize the sum, i.e. $K = \sqrt{L \log L}$. □

### 3.2 CONTINUOUS-FUTUREFILL: QUASILINEAR ONLINE CONVOLUTIONAL PREDICTION

In this section we provide a procedure that significantly improves upon the runtime of Epoched-FutureFill. Our starting point is Proposition 1, which implies that, to compute the convolution between two sequences we can break the sequences at any point, compute the convolution between the corresponding parts and *stitch* them together via a FutureFill computation. This motivates the following Divide and Conquer algorithm to compute the convolution of two sequences $a, b \in \mathbb{R}^L$

- Recursively compute $a_{1:L/2} * b_{1:L/2}$, $a_{L/2+1:t} * b_{1:L/2}$.
- Output the concatenation of $a_{1:L/2} * b_{1:L/2}$ and $(a_{L/2+1:t} * b_{1:L/2}) + \text{FutureFill}(a_{1:L/2}, b)$.

Since FutureFill for $L$ length sequences can be computed in time $O(L \log L)$ via the FFT, it can be seen via the standard complexity calculation for a divide and conquer algorithm that the computational complexity of the above algorithm in total is $O(L \log^2 L)$. As an offline algorithm, this is naturally worse than the computational complexity of FFT itself, however as we show in the following, the advantage of the above algorithm is that it can be executed in an *online* fashion, i.e. the tokens can be generated as the input streams in, with the same computational complexity.

We provide a formal description of the algorithm in Algorithm 2. We note that the formal description of the above algorithm essentially serializes the sequence of operations involved in the above divide and conquer procedure by their chronological order. For high-level intuition we encourage the reader to maintain the divide and conquer structure when understanding the algorithm. The algorithm proceeds as follows: at each time step, $\hat{y}_t = \langle u_{1:t}, \phi_{t:1} \rangle$ is returned as a sum of $C_t$, the cache that stores the contribution from past tokens, and $u_t \cdot \phi_1$, the contribution from token $u_t$. In Line 7, the algorithm then computes the contribution of tokens $u_{t-2^{k(t)}+1:t}$ to positions $t+1, \ldots, t+2^{k(t)}$ of $[u * \phi]$. Finally, we add the output of FutureFill to the existing cache $C$ to accumulate the contributions. In Figure 4, we provide an execution flow for the algorithm for convolving two sequences of length $8$ highlighting each FutureFill operation that is computed.

In the following theorem we prove a running time bound for Algorithm 2. We provide the proof of correctness in the appendix, as it boils down to accounting of contribution from various parts.

**Theorem 3.** *Algorithm 2 computes the online convolution of sequences with length $L$ and runs in total time $O(L \log^2(L))$ with a total additional memory requirement of $O(L)$.*

*Proof.* As can be seen from the algorithm for every generated token the most expensive operation is the FutureFill computed in Line 6 so we bound the total runtime of that operation. Note that at any time $t$, the cost of FutureFill operation is $O((1 \vee k(t)) \cdot 2^{k(t)})$, where $a \vee b$ denotes the max of $a$ and

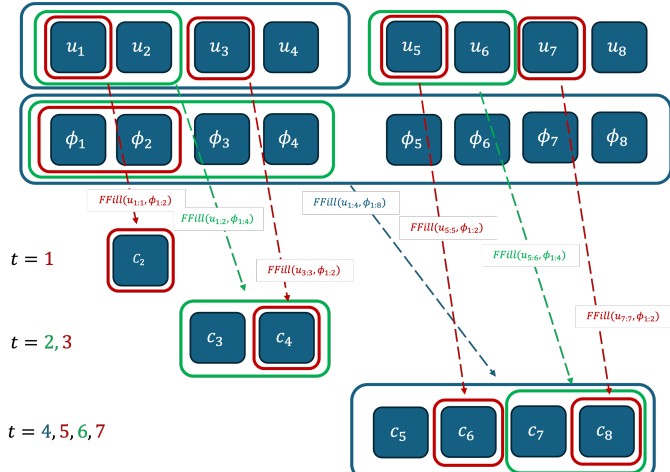

Figure 4: Quasilinear Online Convolution using FutureFill: Figure shows the execution flow for Algorithm 2 for convolving 8-length sequences. The input sequence $u$ streams in an online fashion and the filter $\phi$ is fully available to the algorithm. The colors are representative of the size of the FutureFill operations performed and the time $t$ (also appropriately color-coded) highlights when the FutureFill operations were performed.

b. Summing this over every time step $t$ we get,

$$\sum_{t=1}^{L}(1\vee k(t))2^{k(t)} = \sum_{k=0}^{\lfloor \log L \rfloor}|\{t : k(t) = k\}|(1\vee k)2^k \leq L + \sum_{k=1}^{\lfloor \log L \rfloor} 2^{\lfloor \log L \rfloor - k + 1}\cdot k2^k \leq 3L\sum_{k=1}^{\lfloor \log L \rfloor} k \leq 3L\log^2 L.$$

Thus the total runtime of the algorithm is bounded by $O(L\log^2 L)$.

$\square$

---

**Algorithm 2** Continuous-FutureFill: Quasilinear Generation From Convolutional Models

---

1: **Input:** Convolutional filter $\phi \in \mathbb{R}^L$. Input sequence $u \in \mathbb{R}^L$, streaming one coordinate every round.
2: Set $b = \lfloor \log L \rfloor$. Set FutureFill cache $C \in \mathbb{R}^L$ to 0.
3: **for** $t = 1 \ldots L$ **do**
4:     Receive $u_t$. Output $\hat{y}_t = C_t + u_t \cdot \phi_1$.
5:     Let $k(t)$ be the highest power of 2 that divides $t$, i.e. $k = \max\{i \in [b] : t \mod 2^i = 0\}$.
6:     Compute $\text{FF} = \text{FutureFill}(u_{t-2^{k(t)}+1:t}, \phi_{1:2^{k(t)+1}})$
7:     Set $C_i = C_i + \text{FF}_{i-t} \quad \forall\ i \in [t+1, t+2^{k(t)}]$
8: **end for**
9:

---

## 4   Fast Auto-regressive Sequence Generation from a Prompt

In this section we consider the problem setting of auto-regressively generating $K$ tokens starting from a given prompt of length $L$. For convolutional models specifically we define an abstract version of the problem as follows, given a prompt vector $p \in \mathbb{R}^L$ and a convolutional filter $\phi \in \mathbb{R}^{L+K}$ [4], the aim is to iteratively generate the following sequence of tokens

$$\hat{y}_t = \langle \hat{y}_{1:t-1}, \phi_{t-1:1}\rangle + \langle p_{1:L}, \phi_{t+L-1:t}\rangle = \sum_{j=1}^{t-1}\hat{y}_{t-j}\cdot\phi_j + \sum_{j=t}^{t+L-1}p_{t+L-j}\phi_j.$$

---

[4]the assumption of the filter being larger than $L + K$ is without loss of generality as it can be padded with 0s

As can be seen from the above definition the expected output is an online convolution where the input sequence $u$ has a prefix of the prompt $p$ and the input sequence is appended by the most recently generated output by the model (i.e. auto-regressive generation). We note that we only consider the *convolution* part of a convolutional model (eg. STU) above for brevity and other parts like further projection of the tokens etc can be appropriately added. As mentioned the above model naturally fits into online convolution and the following algorithm delineates the method to use ContinuousFutureFill (Algorithm 2) for the above problem.

---

**Algorithm 3** Fast auto-regressive sequence generation from a prompt using FutureFill

---

1: **Input:** $K > 0, L > 0$, prompt $p_{1:L}$, convolutional filter $\phi \in \mathbb{R}^{L+K}$.
2: Set up a FutureFill cache $C \in \mathbb{R}^K$ as $C \leftarrow \text{FutureFill}(p, \phi)$.
3: Set up the online convolution algorithm (Algorithm 6) with filter $\phi$ and sequence length $K$, i.e. $\mathcal{A} \leftarrow \text{ContinuousFutureFill}(\phi)$.
4: Running candidate token $y \leftarrow 0$.
5: **for** $t = 1, ..., K$ **do**
6:     Output $\hat{y}_t \leftarrow C_t + y$.
7:     Generate next token candidate $y \leftarrow \mathcal{A}(\hat{y}_t)$.
8: **end for**

---

The correctness of the algorithm is immediate via the properties of FutureFill and ContinousFutureFill. The following corollary bounding running time also follows easily from Theorem 3.

**Corollary 4.** *Algorithm 3 when supplied with a prompt of sequence length $L$, generates $K$ tokens in total time $O(L \log L + K \log^2 (L + K))$ using a total cache of size $O(K)$.*

## 5 EXPERIMENTS

In this section, we use a convolutional model that generates tokens in an online fashion to verify our results. We experimentally evaluate Epoched-FutureFill (Algorithm 1) which has a runtime of $O(L^{3/2}\sqrt{\log L})$ and Continuous-FutureFill (Algorithm 2) which has a runtime of $O(L \log^2 L)$ against the naive implementation which has a runtime of $O(L^2)$ when generating $L$ tokens from scratch. For increasing values of $L$, we measure the time $S(L)$ it takes for a single layer to generate $L$ tokens. In Figures 5 and 6 we plot the amortized step time $S(L)/L$ and total generation time $S(L)$, respectively, as functions of $L$. We see the behavior that is expected: the naive decoder runs in amortized $O(L)$ per step, while our methods achieve sublinear and logarithmic decoding complexities respectively.

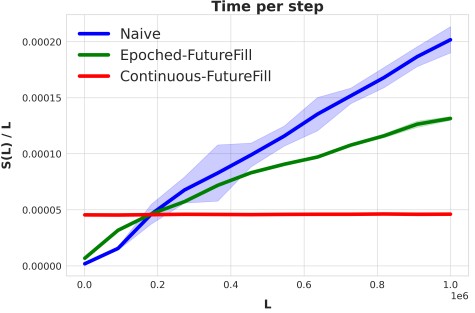

Figure 5: Average number of seconds per step when generating $L$ tokens, as a function of $L$.

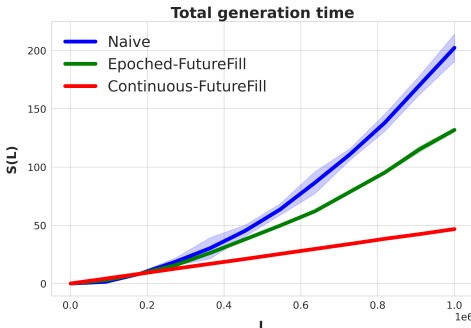

Figure 6: Total number of seconds to generate $L$ tokens, as a function of $L$.

Due to differences in hardware acceleration, inference pipeline implementation, and other engineering details, it would be difficult to present timing results with a properly-optimized setup. On large decoding platforms involving prefill caching, these variations only become more complicated. We opted to time things for one layer on CPU in a simple online decoding loop with a large number of tokens to make the asymptotic gains clear.

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

# A  UPDATED EXPERIMENTS SECTION AND DETAILS

In this section, we employ a convolutional sequence prediction model that generates tokens in an online fashion to verify our results. We experimentally evaluate Epoched-FutureFill (Algorithm 1) which has a runtime of $O(K^{3/2}\sqrt{\log K})$ and Continuous-FutureFill (Algorithm 2) which has a runtime of $O(K \log^2 K)$ against the naive implementation of convolution which has a runtime of $O(K^2)$ when generating $K$ tokens from scratch. We also provide a comparison with a self-attention based Transformer model (with a standard implementation of KV cache and with the same hidden dimension, number of layers and commensurately chosen other parameters, see next subsection for complete details on these models)

For increasing values of $K$, we measure the time it takes for the model to generate $K$ tokens from scratch (i.e. no prompt provided). In Figure 7 we plot the amortized step time the total generation time , as functions of $K$. We see the behavior that is expected: the naive decoder runs in total time $O(K^2)$ per step, similar to the decoder for transformer while our method EpochedFutureFill is able to achieve a significant sub-quadratic improvement.

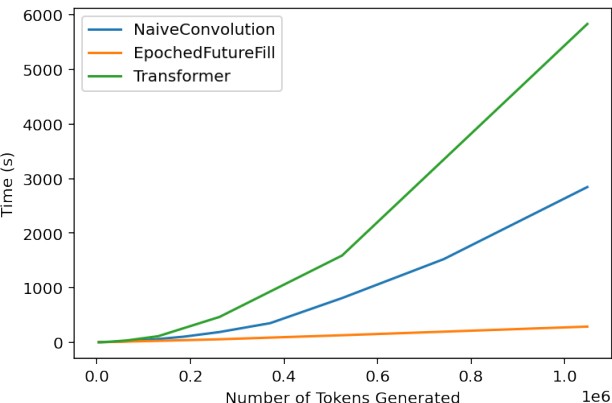

Figure 7: Total time for generating $K$ tokens, as a function of $K$.

## A.1  EXPERIMENT DETAILS

For our experiments we consider a two layer model with either multi-headed self-attention layers (referred to as Transformer) or the STU layers (referred to as convolutional network). The hidden dimension ($d$) of the network is fixed to be 32 with the number of heads fixed to be 4 and the key/value size kept at 8. The networks contain standard implementations of residual connections, layer-norms and a feed-forward (FFN) layer in between every layer. The FFN layer used in the experiments is the FFN$_{\text{GeGLU}}$ layer proposed in Shazeer (2020). For the attention layers we employ a standard implementation of KV-cache for efficiency (i.e. caching the KV values for previously generated token for every layer).

For the STU layer we use the tensored-approximation experimented upon in Liu et al. (2024). Note that during inference for the tensored-approximation of STU, the layer maintains as parameters two matrices $M_{\text{input}} \in \mathbb{R}^{d \times d}$ and $M_{\text{filters}} \in \mathbb{R}^{(K) \times d}$, where $K$ is the number of generated tokens and $d$ is the hidden dimensionality. We provide a detailed equation describing the exact operation performed by the STU layer in generating the $k^{th}$ token below. Let $x_1 \dots x_K$ be the embeddings of tokens generated in online manner, i.e. when generating the $k^{th}$ token only the embeddings $x_1 \dots x_{k-1}$ are available to the model. The generated token sequence follows the following implicit equation

$$[x_k \dots x_1] = \big[ \, M_{\text{filters}} * \big( M_{\text{inputs}}[x_{k-1} \dots x_1] \big) \, \big]_{1:k}.$$

The above convolution operation is over $d-$dimensional sequences which is implemented as $d$ 1-dimensional convolutions performed along each dimension. Since we have equated the hidden dimensionality of the network across all our settings, we can see that, naively computed, the number of flops per token of both the self-attention model as well as the convolutional model are of the

same order which is also observed in the experiments. The next section provides details on how to leverage FutureFill based online convolution algorithms for the STU layer which we implement in our experiments.

Finally all our experiments are implemented in Jax Bradbury et al. (2018) were performed on a single Google TPUv2 machine (Jouppi et al. (2020)).

## B  FAST ONLINE CONVOLUTIONAL PREDICTION

The techniques for online convolution can naturally be applied to online prediction using a convolutional model. We demonstrate this use case via its application to the STU architecture proposed in Agarwal et al. (2023) based on the spectral filtering algorithm (Hazan et al., 2017).

### B.1  CASE STUDY: FAST ONLINE SPECTRAL FILTERING

We illustrate in more detail how the method works for the STU model in Algorithm 4. It improves the total running time from $O(L^2)$ of the original spectral filtering algorithm from Hazan et al. (2017) to $O(L \log^2 L)$ while maintaining the same regret bound.

---

**Algorithm 4** Efficient Spectral Filtering via FutureFill

---

1: **Input:** $K > 0, L > 0$.
2: Set variables $\{M_1^1 \ldots M_K^1 \in \mathbb{R}^{d_{out} \times d_{in}} \leftarrow 0\}$ and set $\{\phi_1 \ldots \phi_K\}$ as the largest eigenvectors of $H_L$, the Hankel matrix corresponding to length-$L$ sequences.
3: Initialize $K$ OnlineConvolution modules, one for each filter $\{\mathcal{A}_k(\phi_k)\}_{k=1}^K$.
4: **for** $t = 1, 2, ..., L$ **do**
5:    Receive input token $u_t$.
6:    **for** $k = 1, 2, \ldots K$ **do**
7:       $F_k \leftarrow \mathcal{A}_k(\phi_k)(u_t)$.
8:    **end for**
9:    Compute and predict $\hat{y}_t = \sum_{k=1}^K M_k^t F_k$.
10:    Observe $y_t$, suffer loss $\ell_t(M_{1:k}^t) = \|y_t - \hat{y}_t\|^2$, and update $M_{1:k}^{t+1} \leftarrow \nabla \ell_t(M_{1:k}^t)$.
11: **end for**

---

The main claim regarding the performance of Algorithm 4 follows directly from Theorems 2 and 3 and is as follows.

**Corollary 5.** *Algorithm 4 with sequence length $L$ guarantees the same regret bound as spectral filtering (Hazan et al., 2017) with context length $L$. Furthermore its computational complexity based on the online convolution module used are as follows:*

- *If using EpochedFutureFill(Algorithm 1): Runtime - $O(L^{3/2}\sqrt{\log L})$, Memory - $O(\sqrt{L \log L})$.*

- *If using ContinuousFutureFill(Algorithm 2): Runtime - $O(L \log^2 L)$, Memory - $O(L)$.*

## C  MISSING PROOFS

*Proof of Proposition 1.* Note that by definition, $[a * b]_s = \sum_{i=1}^s a_i b_{s+1-i}$. We now consider the two cases: for $s \leq t_1$, we have that

$$[a_{1:t_1} * b_{1:t_1}]_s = \sum_{i=1}^s a_i b_{s+1-i} = [a * b]_s.$$

For the case when $t \geq s > t_1$, we have that

$$[a_{t_1+1:t} * b_{1:t-t_1}]_{s-t_1} = \sum_{i=1}^{s-t_1} a_{t_1+i} b_{s-t_1+1-i} = \sum_{i=t_1+1}^s a_i b_{s+1-i},$$

where the last equality follows by redefining $i = t_1 + i$. Further we have that

$$[\text{FutureFill}(a_{1:t_1}, b)]_{s-t_1} = \sum_{i=1}^{t-s+t_1} a_{t_1-i+1} \cdot b_{s-t_1+i} = \sum_{i=1}^{t_1} a_{t_1-i+1} \cdot b_{s-t_1+i} = \sum_{i=1}^{t_1} a_i \cdot b_{s+1-i},$$

where the second last equality follows by noting that $a_j$ is assumed to be 0 for all $j \leq 0$ and the last equality follows by redefining $i = t_1 - i + 1$. Overall putting the two together we get that

$$[a_{t_1+1:t} * b_{1:t-t_1}]_{s-t_1} + [\text{FutureFill}(a_{1:t_1}, b)]_{s-t_1} = \sum_{i=1}^{t_1} a_i \cdot b_{s+1-i} + \sum_{i=1}^{t_1} a_i \cdot b_{s+1-i} = \sum_{i=1}^{s} a_i \cdot b_{s+1-i} = [a*b]_s.$$

This finishes the proof. $\qquad \square$

*Proof of correctness for Algorithm 1.* Consider any time $t$ and the output $\hat{y}_t$. Let $t' \leq t$ be the last time when Line 7 was executed, i.e. FutureFill was computed. By definition $t' = t - \tau$. Note the following computations.

$$\hat{y}_t = \sum_{j=1}^{\tau} u_{t+1-j} \cdot \phi_j + C_\tau = \sum_{j=1}^{\tau} u_{t+1-j} \cdot \phi_j + [\text{FutureFill}(u_{1:t'}, \phi_{1:t'+K})]_\tau$$

$$= \sum_{j=1}^{\tau} u_{t+1-j} \cdot \phi_j + \sum_{j=1}^{t'+K-\tau} u_{t'-j+1} \cdot \phi_{\tau+j}$$

$$= \sum_{j=1}^{\tau} u_{t+1-j} \cdot \phi_j + \sum_{j=1}^{t'} u_{t'-j+1} \cdot \phi_{\tau+j}$$

$$= \sum_{j=1}^{\tau} u_{t+1-j} \cdot \phi_j + \sum_{j=1}^{t-\tau} u_{t-\tau-j+1} \cdot \phi_{\tau+j}$$

$$= \sum_{j=1}^{\tau} u_{t+1-j} \cdot \phi_j + \sum_{j=\tau+1}^{t} u_{t-j+1} \cdot \phi_j = [u * \phi]_t$$

$$\square$$

*Proof of correcteness for Algorithm 2.* We will focus on showing that $C_t = \sum_{i=2}^{t} u_{t+1-i} \phi_i$. Since the output is $C_t + u_t \cdot \phi_1$, this will suffice for the proof. For brevity of the proof and without loss of generality we will assume $L$ is a power of 2. For cleaner presentation for the $s^{th}$ coordinate of vector $v$ we will use the notation $v_s$ and $v[s]$ interchanegably in this section.

We first introduce some definitions for convenience in this section. Given an index $i \leq L$ we define its decomposition $\{i_1, i_2 \ldots i_m\}$ as the unique sequence of numbers $\leq \log L$ such that following holds

$$i_1 > i_2 > i_3 \ldots \text{ and } i = \sum_j 2^{i_j}.$$

These indices correspond to the ones in a $\log L$-bit representation of $i$. Note that $k(i)$ as defined in the algorithm is equal to $i_m$. Further we define the cumulants of $i$ as the following sequence of numbers $\{i'_1, i'_2 \ldots\}$ satisfying

$$i'_\tau = \sum_{j=1}^{\tau} 2^{i_j}.$$

Thus we have that $i'_1 < i'_2 < \ldots i'_m = i$. We now prove the following lemma which specifies when the FutureFill cache gets updated in an execution of the algorithm.

**Lemma 6.** *Given an index $i \leq L$, consider its decomposition $\{i_1, i_2 \ldots i_m\}$ and cumulants $\{i'_1, i'_2 \ldots i'_m\}$ as defined above. It holds that the value of $C_{i+1}$ is updated (as in Line 8 in the algorithm) only when $t$ is one of $\{i'_1, i'_2 \ldots i'_m\}$.*

A direct consequence of the above lemma is that given any index $i$ we have that the value of $C_{i+1}$ is not updated after time step $i$. Further using the decomposition $\{i_1, i_2 \ldots i_m\}$ and cumulants $\{i'_1, i'_2 \ldots i'_m\}$ of $i$ and the update equations for $C$ (Line 8), we have that final value of $C_{i+1}$ is given by the following,

$$C_{i+1} = \sum_{j=1}^{m} \text{FutureFill}(u[i'_j - 2^{i_j} + 1 : i'_j], \phi[1 : 2^{i_j+1}])[i + 1 - i'_j]$$

$$= \sum_{j=1}^{m} \sum_{k=1}^{2^{i_j}} u[i'_j - k + 1] \cdot \phi[i + 1 - i'_j + k]$$

$$= \sum_{j=1}^{m} \sum_{r=i'_j - 2^{i_j}+1}^{i'_j} u[r] \cdot \phi[i + 1 - r + 1]$$

$$= \sum_{r=1}^{i} u[r] \cdot \phi[i + 1 - r + 1]$$

Thus the output of the algorithm for any $i$, satisfies

$$\hat{y}_{i+1} = C_{i+1} + u_{i+1} \cdot \phi_1 = \sum_{r=1}^{i} u[r] \cdot \phi[i+1-r+1] + u_{i+1} \cdot \phi_1 = \sum_{r=1}^{i+1} u[r] \cdot \phi[i+1-r+1] = [u*\phi]_{i+1}.$$

This proves the requisite. We finally provide a proof of Lemma 6 to finish the proof.

*Proof of Lemma 6.* By the definition of the algorithm, to be able to update $C_{i+1}$ at some time $t < i + 1$ it must be the case that

$$i + 1 \in [t + 1, t + 2^{k(t)}].$$

Consider some $t$ and its decomposition $\{t_1, t_2 \ldots t_n\}$ and cumulants $\{t'_1, t'_2 \ldots t'_n\}$. By the definition of the update in Line 8, we have that at time $t$ we only update indices $i + 1$ for which $i$ has the sequence $\{t'_1, t'_2 \ldots t'_{n-1}\}$ in its decomposition as a prefix. It can then be seen that for a given number $i$, the only such numbers are its cumulants, i.e. $\{i'_1 \ldots i'_m\}$ which finishes the proof. $\square$

$\square$

