# OpenReview forum: "FutureFill: Fast Generation from Convolutional Sequence Models"
_ICLR.cc/2025/Conference — Submitted to ICLR 2025_

### Official Review · Reviewer_raFq · 2024-11-02

**Soundness:** 3
**Presentation:** 3
**Contribution:** 2
**Rating:** 5
**Confidence:** 4

**Summary:**

The paper introduces a new method to autoregressively generate tokens from a convolutional model, improving the number of flops required to generate K tokens, both from scratch and with a prompt of length L. The core idea is to use a precomputed cache to avoid repeated computation of the same quantities. The cache is also smaller than that of alternative approaches.

The theoretical analysis is confirmed with an synthetic empirical study.

**Strengths:**

The algorithms are presented are simple and it’s clear how the required number of FLOPs is improved when using the algorithm. The improvement in the number of required FLOPs is significant.

With a practical implementation showcasing speedups over standard autoregressive generation for convolution models, the algorithm could have a good impact on the community.

**Weaknesses:**

The biggest weakness is that the authors don't show that the algorithm gives practical speedups on real tasks of interests, such as language modeling.

Even if the algorithm is not giving improvements on practical tasks it would have been useful to understand why it does not work well in these settings, so someone else can more easily build on top on the work.

The experimental section is also unclear in it's current form, as it's not stated what operation S(T) implements in detail.

The novelty of Algorithm 1 is also limited.

**Questions:**

What is S(T)? Is it just a convolution operation or a full convolutional layer?

Did you try the algorithm on any real tasks?

In figure 4, the improvement over the naive algorithm to generate 5e5 tokens is less than 2x. Given the difference in the theoretical number of FLOPs I would have expected a larger difference. Why is this?

Can you expand the Figure 4 to generate a larger number of tokens so that the difference between the suggested approach and the naive approach is more clear.

---

> ### Author Response · Authors · 2024-11-23
> **Thanks for your review**
>
> We thank the reviewer for the review and appreciation of the strengths of the paper. Regarding the point of demonstrating practical speedups we refer the reviewer to the general response to put our contribution in the right light as well as the additional experiments we have provided in the Appendix (Section A) in the revised paper. We believe these experiments are close to a practical setting in language modelling and we do indeed see a significant improvement of our method. We answer the specific questions here –
>
> - In our newly provided set of experiments we fully detail the operations measured. They now represent a fully formed two layer language model with STU convolutional layers. We note that the task in our  newly uploaded experiments is widely used, i.e. decoding of language models. The main variations one would find in real tasks here are changes of the model specific parameters like number of hidden dimensions, number of layers etc.
>
> - In the new experiments, we have provided a large horizon on the number of tokens generated and we do indeed see significantly larger gains as well as get a sense of the trends and scaling.

---

> > ### Comment · Reviewer_raFq · 2024-12-02
> > **Official Comment by raFQ**
> >
> > Thank you for the response. I have a few questions regarding the updated experiment section.
> >
> > 1. What does the light blue shaded region represent in figure 5 and 6? How is it calculated?
> > 2. Figure 4 in the previous version of the paper suggested that FutureFill was faster than the naive method for 1e5 steps but Figure 6 in the new version suggest that the naive version is faster than FutureFill for 1e5 steps. Why is this the case?

---

> ### Author Response · Authors · 2024-12-02
>
> Thank you for your questions!
>
> 1. The results in Figures 5 and 6 are from averaging the timing experiments over 16 trials. The shaded regions represent + or - 1.96 times the standard deviation of the timing measurements for each method and number of decoding tokens (ie 95% confidence intervals). In our experiments, only for the naive decoding method is the variation noticeable enough to be seen on the graph, though this detail may be different with pre-compilation or hardware acceleration.
> 2. In order to add the continuous futurefill experiments, between the initial submission and the revision we switched the setup to use a numpy implementation instead of a jax one (which originally was implemented with a jax.lax.scan loop). There are extra considerations for a jax implementation of continuous futurefill (such as bottoming out the recursion to avoid creating kernels of all sizes, as mentioned to another reviewer) that made it more straightforward to do the timing in basic numpy. This difference in implementation leads to differences in the exact runtimes and the constant factors (and the difference at 1e5 steps you observe), but has the same asymptotic behaviors. We should have specified this difference in the revised version, we apologize. Thank you for pointing it out!
>
> We remark that because these are unoptimized implementations for illustrative purposes, the exact times for particular decoding lengths in figures 5 and 6 are probably not exactly what would be seen when deploying our methods in serious applications. However, (as demonstrated in Appendix A) optimized implementations on accelerated hardware will also have immediate benefit over the naive decoding. The algorithmic advantage our methods provide is clear, present in both synthetic and accelerated settings, and improves with scale (which is what we set out to prove in this paper) — any airtight implementation (such as what would be developed for an industry application, for example) would likely require low-level design decisions specific to the application. For universality and clear presentation, we opted for simplicity over complicated optimality in our experiments. We hope this answers your questions and provides some context for our experimental choices!

---

> > ### Comment · Reviewer_raFq · 2024-12-02
> > **Official Comment by raFQ**
> >
> > Thank you for the response. I will maintain my current score.

---

### Official Review · Reviewer_SQi8 · 2024-11-03

**Soundness:** 2
**Presentation:** 2
**Contribution:** 2
**Rating:** 3
**Confidence:** 2

**Summary:**

The paper proposed an acceleration method for convolutional models called FutureFill, which reduced the asymptotic computation time of convolution models to $O(K\sqrt{L\log L})$ and $O(L\log L +K^2)$ in two respective settings and reduced cache size to $O(K)$ for generation from scratch. The theoretical acceleration was verified by experimental results.

**Strengths:**

- The proposed acceleration method reduces the generation time and cache size, which is said in the paper to be applicable to many types of convolutional models.

**Weaknesses:**

- Limited evaluation.
	- Unclear setting. The evaluation (section 4.2) is missing information on detailed model architecture, size, timed operators, distribution of L and K, or hardware details, making the results hard to interpret or reproduce.
	- Limited results. Only a few data points from a single model are plotted. It would be more solid to cover results for models mentioned to be applicable in section 2.1 in a realistic setting to verify the end-to-end speedup and saved cache memory.
	- Experimental results for Algorithm1 compared with previous methods are not included.

- Limited novelty and contribution. The major improvement for Algorithm1 compared with [1] appears to be the change of convolutional caches, and the improvement in asymptotic time from  $O(L(\log L +K) +K^2)$ to $O(L\log L +K^2)$ is not significant.

Minor:
- a typo in line 66. $K\sqrt{L}\log L \to K\sqrt{L\log L}$

---

References:

[1] Laughing Hyena Distillery: Extracting Compact Recurrences From Convolutions http://arxiv.org/abs/2310.18780

**Questions:**

- Can you provide more details on the evaluation setting?

- Can you provide additional results on more models mentioned in section 2.1 with a setting closer to realistic deployment?

---

> ### Author Response · Authors · 2024-11-23
> **Thanks for your review**
>
> We thank the reviewer for their review. We refer the reviewer to our general response regarding further comparisons. We disagree with the reviewer on the point of Limited Novelty and Contribution. Computing convolutions in an online fashion (such as the one during sequence inference) is a fundamental problem and no general improvement was known beforehand. Our method provides such an improvement. We believe this is a fundamental algorithmic improvement to a fundamental problem with a potential for wide-ranging impact. We respectfully disagree with the following comment by the reviewer
>
> *The major improvement for Algorithm1 compared with [1] appears to be the change of convolutional caches, and the improvement in asymptotic time from $O(L(\log⁡L+K)+K^2)$ to $O(Llog⁡L+K^2)$ is not significant.*
>
> It is absolutely clear in the setting that this result is reported i.e. $K << L$,  $K \cdot L$ would be the dominant term in the known implementation and removing this term is clearly a significant improvement as acknowledged by other reviewers. If the reviewer typically considers the setting $K \sim L$ then as clearly stated in the paper they would use the second algorithm which would give them a complexity of $(K+L)\sqrt{(K+L)}$ which is also a clear and significant improvement over $(K+L)^2$ (the best known).

---

### Official Review · Reviewer_hM23 · 2024-11-03

**Soundness:** 3
**Presentation:** 2
**Contribution:** 2
**Rating:** 5
**Confidence:** 4

**Summary:**

Proposes a future fill algorithm for prompt-based and online inference of convolutional-based sequence models. In the first setting, authors use FFT to precompute a cache to generate the next K tokens given L tokens. In the second setting, they generate from scratch and create partial caches, to construct the output at the end of each token iteration. In both settings, authors prove their claimed asymptotic bound reductions.

**Strengths:**

This paper attempts to address a relevant problem involving reducing inference's quadratic complexity.

Neat setup of the mathematical framework and proof of reducing the asymptotic complexity of setting 1 to O(L L+ K^2 ) and second setting to  Ksquare_root( L log L) .

The application of these techniques to spectral filtering is explained well.

**Weaknesses:**

In the first setting of generation with prompt, the assumption that K is usually less than L is only applicable for a subset of tasks such as summarization.

The experimental analysis is insufficient in terms of details like hardware used, implementation framework, proof of correct implementation, timing measurements, models used, time breakdown, memory analysis, input data sizes, among others.

**Questions:**

1. Since GPUs are typically used for inference of LMs, showing the gains in end-to-end inference time and how well the reduced asymptotic complexity translates to real performance gains needs to be validated.

2. The algorithms improve the complexity of convolutions, but models usually have several other components like projections, gating, MLPs, etc.
 2a. It would be valuable to gauge the contribution of each of these components to final inference times.
 2b. How different parameters like the number of layers, hidden dimension size, scaling with sequence lengths, batch size, etc affect the performance of the proposed algorithms.

3. Experimental analysis would benefit from a deeper analysis and also presentation.

---

> ### Author Response · Authors · 2024-11-23
> **Thank you for your review**
>
> We thank the reviewer for their review. Regarding further comparisons we would like to refer the reviewer to the central response where we cover many of the questions raised by the reviewer. Regarding the point of GPU validation a similar point was raised by another reviewer which we repeat here –
>
> - Applicability to GPUs:
>
> Regarding the use of GPUs without optimized FFT support, we acknowledge the ongoing efforts to improve FFT performance on GPUs, including recent works such as Flash FFT Convolution (https://arxiv.org/abs/2311.05908) and earlier studies (https://arxiv.org/abs/2302.06646). These works demonstrate that FFT-based convolutional methods now outperform traditional attention mechanisms on GPUs due to their faster computational complexity. We would like to draw the reviewers’ attention to Table 6 in Flash FFT Convolution paper ((https://arxiv.org/abs/2311.05908)) and Figure 1 in the Laughing Hyena paper (https://arxiv.org/abs/2310.18780), which show that FFT-based convolution methods significantly outperform attention in GPU settings.
>
> In our practice we have found even a simple implementation of Bailey’s FFT (https://en.wikipedia.org/wiki/Bailey%27s_FFT_algorithm) which leverages simple matrix multiplications is already sufficient to be used in our experiments on GPUs/TPUs and provides the benefits reported in our experiments. We note that our newly uploaded experiments are on a Google TPUv2 and indeed written in a way to leverage accelerators for all the compared models.
>
> - Other concerns:
>
> *In the first setting of generation with prompt, the assumption that K is usually less than L is only applicable for a subset of tasks such as summarization.*
>
> Please note that we have provided two algorithms – one dealing with the setting of prompt + generation wherein we make the assumption of $K < L$. Furthermore we also provide an algorithm wherein we provide an algorithm for continuous auto-regressive generation. In the setting where $K > L$ we would apply the latter algorithm and the generation complexity per token would be $O(\sqrt(K + L))$ which is an improvement over the naive algorithm of $O(K+L)$.
>
> *The algorithms improve the complexity of convolutions, but models usually have several other components like projections, gating, MLPs, etc. 2a. It would be valuable to gauge the contribution of each of these components to final inference times. 2b. How different parameters like the number of layers, hidden dimension size, scaling with sequence lengths, batch size, etc affect the performance of the proposed algorithms.*
>
> Upon the reviewer’s requests in the updated version (Section A in the appendix) we have provided a comparison of the various models with a 2-layer set up with projections, feed-forward layers etc to get a sense of the contribution to inference times of those components. We would like to note (and as can be seen in the experiment provided) that scaling with sequence length (i.e. the number of generated tokens) is in our favor as the other components contribute a constant amount in terms of runtime per generated token. As the sequence length scales the effect of these constants vanish and our algorithm’s improvement in terms of sequence length shines. We would like to note that the performance will naturally scale in our favor as the number of layers or hidden model dimensions increases as there are more convolutions performed in the network in more dimensions. These parameters naturally scale the runtime linearly for every model involved. We have provided all experiment details for the reader in section A.1.

---

> > ### Comment · Reviewer_hM23 · 2024-11-27
> >
> > Thank you for the response.
> >
> > Regarding K less than L: the answer makes sense. Thank you.
> >
> > Regarding GPUs: While the response makes sense logically speaking, actually translating an algorithm to an implementation that utilizes hardware properties can still potentially reveal issues. For example, one of the issues that comes out from your description is that computing the output for partial sequences will require launching several kernels of very small size. This may result in a significant CPU overhead due to the kernel launch latencies. Thus, validating in realistic implementation and experimental settings is essential for this type of work.
> >
> > Regarding the TPU experiment: Thank you for the new experiment. While this validates to some degree the asymptotic claims, how much real speed up this is going to get is not clear given the restricted experimental setup. Specifically, using only 2 layers and an inner dimension size of 32 does not stress the algorithm in terms of saturating compute or memory and does not represent real-world model settings. Thus, experimenting with a much more diverse setup is necessary to see the complete potential in addition to studying additional crucial metrics such as TPU utilization, memory access etc.

---

> > > ### Author Response · Authors · 2024-11-27
> > >
> > > Thank you for your response.
> > >
> > > Regarding GPUs -- please note that our newly uploaded experiments are on TPUs with appropriately Jitted functions. So our implementations utilizes 'hardware properties'. We agree that there is potential for the ContinuousFutureFill (proposed in the revision) to launch several kernels of small sizes which could be potentially problematic. We have the following response
> > >
> > > -- Firstly please note that this issue is not present with EpochedFutureFill (our originally proposed method). To remind the reviewer, this method performs periodic FFT based convolutions to compute a FutureFill for fixed length and then a naive online convolution in each epoch. Note that for EpochedFutureFill the convolution operation is only called for fixed length arguments of size $K$ where $K$ is the total generation length. This is assuming, as is typical also for Attention for efficiency, that one would pre-allocate the entire generation length and generate kernels of that size.  Indeed as we highlight in our TPU experiment the EpochedFutureFill, taking all these points into consideration still outperforms Attention and Naive Convolution by a large margin.
> > >
> > > -- Secondly if speaking about ContinuousFutureFill, we expect practical implementations of ContinuousFutureFill to *bottom the recursion* out before the sequence length reaches as low as 1 for efficiency. This would reduce the number of differently sized kernels for convolution required. Nevertheless, between EpochedFutureFill and ContinousFutureFill, it is definitely unclear practically which algorithm would be a winner and we expect in most practical settings EpochedFutureFill to be a winner which is why in the TPU experiments we have experimented only with that algorithm. We do believe however that it is important for the community to know about ContinuousFutureFill since it nearly achieves the theoretical limit for the problem and hence might be a crucial algorithm especially as hardware evolves.
> > >
> > > -- Regarding an even more *practical* setting, as we have noted previously we expect the gains to increase as layers and d_model increases. We will attempt to run an even more practical setting at the reviewers' request, i.e. either the 117M/417M parameter model presented in Gopher series of models (https://arxiv.org/pdf/2112.11446) (these have 12 layers and 768/1536 model dimensions) and report back. We can only hope that the reviewer is perhaps satisfied by this scale as this is unfortunately the maximum we can attempt to run given the time left.

---

> > > > ### Author Response · Authors · 2024-11-30
> > > > **Results on a larger model**
> > > >
> > > > Here are the results on a larger model. The setup is the same as the Gopher 417M model from this paper -- https://arxiv.org/pdf/2112.11446. Its a 12 layer model with a d_model of 1536 (12 heads, 128 key/value size). The below provides the runtime for generating the listed number of tokens and compares a transformer model, an STU model with naive convolution and an STU model with EpochedFutureFill.
> > > >
> > > > | Algorithm \ Num Tokens Generated | 1024 | 2048 | 4096 | 8192 | 16384 | 32768 | 65536 |
> > > > | --- | --- | --- | --- | --- | --- | --- | --- |
> > > > | Transformer | 9.01 | 17.86 | 42.23 | 124.57 | 415.05 | OOM | OOM |
> > > > | Naive Convolution (STU) | 7.02 | 12.03 | 26.9 | 71.05 | 221.13 | 764.07 | OOM |
> > > > | Epoched FutureFill (STU) | 6.99 | 11.92 | 19.41 | 39.36 | 85.26 | 208.56 | 495.9 |
> > > >
> > > > *OOM - refers to out of memory

---

### Official Review · Reviewer_7b51 · 2024-11-04

**Soundness:** 2
**Presentation:** 1
**Contribution:** 2
**Rating:** 5
**Confidence:** 3

**Summary:**

The paper proposes FutureFill, a fast generation method designed for convolutional sequence models, aiming to reduce generation time complexity from linear to square root relative to context length. FutureFill utilizes a prefill cache that is theoretically more efficient in both time and memory, requiring less storage than traditional convolutional and attention-based models. The authors validate their approach through theoretical analysis and experimental evaluation on synthetic generation tasks.

**Strengths:**

1. FutureFill offers a novel approach to reducing the generation time for convolutional models from linear to square root relative to context length. This theoretical improvement addresses one of the core challenges of convolutional models, positioning FutureFill as a potentially valuable contribution to the field of efficient sequence generation.
2. By using a prefill cache that scales with the number of generated tokens rather than the full context, FutureFill significantly reduces memory overhead. This is particularly valuable in applications where memory is a bottleneck, making the method more accessible for deployment in resource-constrained environments.
3. FutureFill offers an alternative to attention-based models, which suffer from quadratic complexity, especially with longer sequences. By exploring a convolution-based approach, the paper contributes to the broader goal of overcoming the limitations of self-attention in handling long sequences.

**Weaknesses:**

1. The paper relies heavily on synthetic tasks for evaluation, which may not fully represent real-world applications. Without testing FutureFill on more practical benchmarks or large-scale NLP tasks, it’s difficult to assess its true effectiveness and scalability.
2. While the authors emphasize efficiency improvements, they do not compare FutureFill against other advanced, optimized generation techniques.
3. The authors focus mainly on theoretical comparisons with standard convolution and attention models. However, many recent techniques in sequence modeling—such as token pruning, cache compression, or sparse attention mechanisms—are entirely overlooked, which limits the relevance of FutureFill in the context of state-of-the-art sequence generation methods.

**Questions:**

1. How does FutureFill perform on standard language modeling or machine translation benchmarks compared to other optimized sequence models? Including such comparisons would significantly strengthen the paper’s claims about practical applicability.
2. The paper briefly mentions the complexity of benchmarking FutureFill on accelerated hardware, but how feasible is it to implement this method in practice? Given the reliance on FFT, how does FutureFill perform on GPUs without optimized FFT support?
3. The magin formatting of this paper is changed, which may violate the requirement of ICLR.

---

> ### Author Response · Authors · 2024-11-23
> **Thank for your review**
>
> We thank the reviewer for their comments. We request the reviewer to also read the General Response provided by us to all reviewers for more context on comparisons and real-world applicability of our method. In the following we respond to other questions raised by the reviewer.
>
> - Inference Efficiency and Comparison with Approximate Methods:
>
> While comparisons to token pruning, cache compression, and sparse attention mechanisms are valuable, we believe they are outside the scope of this paper. These are methods to approximate the full attention model with performance vs. speed tradeoffs, while we focus on an exact inference method for convolutional sequence models without quality loss. In our view, comparing with these methods requires  understanding and defining the acceptable quality loss inherent in approximate attention models. Many of these methods involve trade-offs between speed and accuracy, often with parameters adjusted on a case-by-case basis. Given this, a fair comparison is difficult to make without a clear understanding of the quality loss in these approximations. As such, we consider it premature to make these comparisons in the current context.
>
> As an example, consider the evaluation style in the Laughing Hyena paper (NeurIPS 2023), which contrasts the approximate Hyena model with the base Hyena model in terms of inference speed and approximation loss. Like us, they do not compare their method against sparse attention models. It is crucial to note that our method does not need to rely on providing trade-offs—our approach provides exact computations but at a significantly faster inference speed, with no need to introduce any approximation or quality loss.
>
> - Applicability to GPUs Without Optimized FFT Support:
>
> Regarding the use of GPUs without optimized FFT support, we acknowledge the ongoing efforts to improve FFT performance on GPUs, including recent works such as Flash FFT Convolution (https://arxiv.org/abs/2311.05908) and earlier studies (https://arxiv.org/abs/2302.06646). These works demonstrate that FFT-based convolutional methods now outperform traditional attention mechanisms on GPUs due to their faster computational complexity. We would like to draw the reviewers’ attention to Table 6 in Flash FFT Convolution paper ((https://arxiv.org/abs/2311.05908)) and Figure 1 in the Laughing Hyena paper (https://arxiv.org/abs/2310.18780), which show that FFT-based convolution methods significantly outperform attention in GPU settings. Moreover, in token generation tasks starting from a prompt, our method can directly leverage these GPU-optimized algorithms, as a single FFT call produces all the necessary prefil cache information for our approach, further boosting performance.
>
> - Our apologies regarding the margin issue. We will fix and correct this.

---

### Author Response · Authors · 2024-11-23
**General Response**

We thank the reviewers for their reviews and evaluation. Multiple reviewers have asked for further evaluations and have raised concerns regarding  the real-world  applicability of our method. In this general response we reiterate the broad ideas which form the basis for our method and clarify why we believe the applicability of our proposed method is in our opinion immediate. We have further conducted upon the reviewers’ requests a comparison between a fully setup multi-layer attention model, convolutional model (STU) with the naive convolution and our proposed method. We have uploaded  these new experiments in the updated paper on the system as the first section of the Appendix. We have also appended  an Experiment Details section with all the details of the experimental setup.

We would like to emphasize that our work presents a general method for exact computation in convolutional models during inference, achieving a per-token complexity of $O(\sqrt{L})$. This is a significant improvement over the previously best-known result of $O(L)$. Prior to our work, it was not clear whether such an improvement was even feasible. As the recent work on the Laughing-Hyena model (see https://arxiv.org/abs/2310.18780) demonstrates, convolutional models naively have the same scaling on inference times as attention, namely $O(L)$. To address this, they resorted to approximating the convolutional model through distillation to an RNN, which sacrifices accuracy for speed. We note that the loss incurred by such approximations is difficult to predict in future settings. Importantly, our paper completely eliminates the need for such approximations, providing an efficient and completely accurate alternative.

The improvement afforded by our method FutureFill, is immediately applicable to any large language model (LLM) based on convolutional architectures. We believe that the reduction in inference time during token generation  is immediate from the design of our method since it performs a subset of the computational operations performed by the naive implementation. Therefore, we consider it self-evident that our method outperforms the naive approach in terms of computational efficiency, even perhaps without the need for experimental validation. While we have added further verification to the paper upon the reviewers’ requests, we note that the exact improvement will vary across different models based on the other non-convolutional components present in the model. Nevertheless it is clear that every model based on convolutions should choose to employ this method as opposed to the naive algorithm as it comes with no loss and a potential for a large gain.

We would like to remind the committee that the improvement we provide is a really fundamental improvement to the natural and widely applicable problem of inference from convolutional sequence models. This is unlike inference improvements through approximation in attention models which come at different costs. Our improvement in comparison is immediate and we believe should be promptly adopted in any convolutional model of which there are several variants being researched in the sequence prediction and modeling community. As a result we request the committee to view the result which we believe to be of immediate and wide impact in the research community in the appropriate  light.

We would further like to bring to the reviewing committee’s attention that in our updated manuscript, using a straightforward  application of FutureFill we have been able to improve the inference speed of our algorithm from $O(\sqrt{L}) \rightarrow O(\log^2(L))$, per token generated, i.e. nearly constant per token closely matching that of RNNs/SSMs. We call the originally proposed method EpochedFutureFill in the revision which achieves $O(\sqrt{L})$ and the new improvement ContinuousFutureFill which achieves $O(\log^2(L))$. We leave it up to the reviewing committee to decide whether to take this significant improvement into account during their evaluation. We are happy to provide further details in comments, should the reviewing committee ask for it.

---

### Author Response · Authors · 2024-11-26
**No engagement thus far**

We thank the reviewing committee for their efforts on this paper. Unfortunately we have not heard back from any reviewer post our rebuttal. We have clarified several of the reviewer concerns, added new experiments and added a new significantly improved method. If there are lingering concerns we would appreciate it if the reviewers let us know so that we can clarify. We believe we have a general method that has wide and immediate applicability for anyone who is looking to build new architectures for LLMs via convolutions and we hope for more engagement from the reviewers for us to clarify any lingering concerns.

---

### Meta-Review · Area_Chair_fE9A · 2024-12-14

**Metareview:**

This paper proposes FutureFill, a fast generation method designed for convolutional sequence models. After rebuttal, it received scores of 3555. One major concern shared by all the reviewers still remains, that is, limited evaluation. In the original paper, the theoretical analysis is confirmed with only a synthetic empirical study. The proposed method is highly suggested to test on real tasks of interests, such as language modeling. During rebuttal, the authors added some new experiments. However, overall, it's still lightweight and insufficient to convince reviewers. Therefore, the AC would like to recommend rejection of the paper.

**Additional Comments On Reviewer Discussion:**

The major concern of the paper is the limited evaluation. The paper relies heavily on synthetic tasks for evaluation, which may not fully represent real-world applications. During rebuttal, the authors have conducted some more experiments in Appendix A, and provided more experimental analysis as requested by reviewers. However, from the reviewer discussions, the added experiments and analysis are still lightweight, therefore not convincing enough for reviewers to increase the scores.

---

### Decision · Program_Chairs · 2025-01-22

Reject